# Potential Possibilities of Using Groundwater for Crop Irrigation in the Context of Climate Change

Ireneusz Cymes [1,*], Ewa Dragańska [1] and Zbigniew Brodziński [2]

1   Department of Water Management and Climatology, University of Warmia and Mazury in Olsztyn, pl. Lodzki 2, 10-708 Olsztyn, Poland; ewad@uwm.edu.pl
2   Department of Economy Competitiveness, Faculty of Economic Science, University of Warmia and Mazury in Olsztyn, 10-719 Olsztyn, Poland; zbr@uwm.edu.pl
*   Correspondence: irecym@uwm.edu.pl

**Abstract:** The study analyzed the structure of water shortages in plant crops and the available groundwater resources that can be used to satisfy these needs. The research was carried out in Braniewo poviat, which can be considered representative of the conditions of Central and Eastern Europe. A clear upward trend in the temperature value was observed, which influenced the changes in the duration of thermal seasons and agricultural periods. It also increases the intensity of the evapotranspiration process, which results in the reduction of water resources. The presence of significant water shortages, especially in the cultivation of root crops, such as, for example, late potato or sugar beet, justifies the need to irrigate these plants. Due to unevenly distributed surface water resources, groundwater is used as a source of irrigation. It was found that in the case of many crops, the areas with the greatest water shortages were those with average or high abundance in available groundwater. When indicating the possibility of abstracting large amounts of groundwater for use in plant production in Braniewo poviat, one should consider the fact that, in the long term, their exploitation may cause negative environmental effects.

**Keywords:** climate change; water shortages for agricultural crops; irrigation; groundwater resources





## 1. Introduction

The causes and scenarios of climate change, their consequences and adaptation processes have been described in subsequent IPCC reports [1]. The latest report "Climate Change 2022: Consequences, Adaptation and Risks" highlights the problems associated with the already observed increase in extreme weather events such as extreme high and low temperatures, prolonged droughts, or rapid precipitation, and problems with the availability of drinking water [2]. A major cause of the increasing frequency of extreme weather events in many parts of the world [3–5] is rising air temperatures [4–6]. This trend is projected to continue in the coming decades unless appropriate action is taken [4,6–8], as the accumulation of greenhouse gases mainly from human activities increases [1–3].

It is estimated that by 2025, every fourth person on Earth may suffer from extreme water scarcity [2]. Central Europe, including Poland, lies in a temperate climate in which the effects of drought are often downplayed [9,10]. However, as observed, with global climate change, the problem of drought, both nationally and regionally, is becoming more serious [11]. Meteorological data show that even the northern areas of Central Europe may be threatened by prolonged periods without the occurrence of sufficient precipitation to meet the needs of plants, as observed in 2018–2020 [12]. Studies on climate change in Poland and its impact on agriculture clearly show that extreme weather, droughts and heat waves, as well as heavy precipitation, will have a decisive impact on the growth, development and yield of plants. The predicted warmer climate significantly affects the planning of agricultural practices, accelerating sowing and harvesting dates across the

country. Research indicates that soil moisture anomalies (deficits or surpluses) are a key factor in agricultural yield increases [13–18].

Changes in precipitation distribution are also becoming more pronounced, with precipitation increasing in winter and decreasing in spring and summer. In addition, the intensity of precipitation intensifies, making its use by plants less efficient [5]. Increasing air temperature, combined with limited water availability during the growing season, leads to an imbalance between precipitation and plant water needs. Extreme weather events resulting from climate change are responsible for yield fluctuations ranging from 18 to 43 percent [19]. This situation affects the yield and quality of agricultural products and consequently global food security [20–24]. The example of Spain shows that the demand for irrigation water by year 2100, depending on the type of crop, will increase by 40–250% [25]. In Cyprus, a typical water-poor country with the highest water stress index among European countries, it is estimated that in the period 2031–2060 the average annual rainfall will decrease by 5 to 15% [26]. In many parts of the world, evapotranspiration is projected to increase by 25% by the year 2080, resulting in an increase in total irrigation water requirements [27,28].

According to the Food and Agriculture Organization of the United Nations (FAO), irrigation of crops uses 70% of the available freshwater [29,30]. Therefore, efficient use of water for irrigation and increased efficiency of irrigation systems are key factors enabling sustainable water management and adaptation to climate change [31–34]. The practice of assessing the effects of different agricultural practices on water yield (i.e., "more yields per drop") is no longer an academic invention but is of great practical importance [12]. The initiatives taken to improve the efficiency of water use have become necessary, especially in those areas where water is the most limiting factor in plant production [5]. Water use efficiency (WUE) and water productivity (WP) indicators are widely used in the assessment of water consumption in agricultural production systems [12]. WUE refers to the ratio (or percentage) of water that is productively used by a plant, while WP refers to the ratio of production to water used [35]. Moreover, the WP index is often used to construct measures characterizing the ratio of, for example, yield, obtained biomass, or their economic value to, e.g., transpiration, evapotranspiration, irrigation, water supply, etc. [36]. To estimate farm-scale water use, the FAO has developed guidelines to help calculate agricultural water optimization indicators. These new tools are designed to implement resource-efficient agronomic practices and improve farm management [35,37].

In order to obtain optimal yields with limited water resources, farmers must possess new technologies and varieties of crops that will guarantee more efficient water management [38]. With this goal in mind, water networks and irrigation systems have been modernized in many countries. In the case of irrigation systems, gravity irrigation is abandoned in favor of drip irrigation because this change alone gives up to 90% water savings [39]. In less developed countries, in order to reduce poverty, the World Bank supports the use of improved irrigation systems. In the countries of the European Union, the improvement of irrigation systems is one of the priority actions defined in the European Water Framework Directive [40]. The above-mentioned directional changes, apart from the obvious benefits, also bring negative effects. Namely, the pumps used in modern irrigation systems require the supply of a significant amount of electricity, which makes electricity another resource crucial for the functioning of irrigation. Energy expenditure on supplying water to irrigation networks and systems increases the carbon footprint [41]. One of the most interesting proposals to reduce the energy consumption of water abstraction from wells is based on the installation of variable speed pump drives, which allow for potential energy savings of up to 23% with payback periods from 4.5 to 10 years [42]. Renewable energy sources, such as photovoltaic energy, wind energy, or energy recovery from overpressure, will play an important role in the water supply [41]. Despite the high awareness among decision makers regarding the need to modernize drip irrigation and sprinkler irrigation systems, there are still barriers that make farmers not interested in using such solutions. The problem mainly concerns the lack of investments related to the construction

of the network, little support, lack of trained personnel to operate the equipment and the risk associated with the purchase of equipment that is not widely used today [31,43–45].

Significant water savings can be achieved through the use of improved irrigation systems, but in some cases, this may be controversial in the long term [21]. It is often argued that increasing the efficiency of the use of irrigation systems will result in an increase in the irrigated area, which will significantly reduce the process of recharging aquifers [46]. Moreover, excessive irrigation of crops may lead to nutrient leaching from the soil and, consequently, groundwater contamination [47,48]. It may also limit the root aeration process and promote the multiplication of pathogens [49]. Importantly, accessibility to groundwater and surface water may also be limited during long-term and successive dry periods. Therefore, it can be expected that conflicts will increase regarding the possibility of exploiting limited surface water resources and lowering the groundwater table [48]. Another important factor is the increase in energy consumption and the increase in carbon dioxide emissions as a result of the use of pumps for extraction of groundwater, pumping it and distributing it over large areas [50]. An example of irrigation use in Mediterranean countries shows that there is often an increase in salinity and soil degradation [24,51]. It is estimated that worldwide, by 2050, more than 50% of arable land will have soil quality problems. Already, around 10 million hectares are abandoned each year due to soil salinity [52]. The degree of soil salinity is closely related to climatic conditions and the presence of a source of readily soluble salts. The problem concerns irrigated soils of dry climates of steppes, semi-deserts and deserts, where evaporation prevails over water absorption. Under the conditions of the Polish climate, if the irrigation of plants is strictly connected with their proper fertilization, the danger of salinization of soils under the influence of irrigation does not occur. Salinization also occurs in the case of soils located within the range of influence of sea waters, soils irrigated with sewage and organic fertilizers with high salt concentrations, and soils exposed to saline inflow from anthropogenic sources [53]. These experiences from other countries show that the use of groundwater for irrigation of agricultural crops requires special care. The renewal time for these resources is much longer than for the use of surface water. The development of irrigation systems for crops and the related increase in the density of the number of wells may consequently have an irreversible, negative impact on the environment.

The aim of the research was to determine, in the conditions of Braniewo poviat, the structure of the occurrence of available groundwater resources and, on this basis, to assess the possibility of covering the water shortages in cultivated plants with these resources.

## 2. Study Area

The area covered by the research is located in north-eastern Poland, in Warmińsko-Mazurskie Voivodeship. Braniewo Poviat borders with the Russian Federation from the north, and from the north-west, its natural border is Vistula Lagoon (Figure 1). According to the physical and geographic division [54], the area is located on the border of two provinces: the Central European Lowlands and the East Baltic-Belarusian Lowlands.

The climate in this region has both continental and marine features. It is specific due to the fact that it is characterized by the lowest amplitudes of average annual temperatures in Poland, the occurrence of strong winds and a short and mild winter. The continental features of the climate are mitigated by the vicinity of the Baltic Sea. The number of days with snow cover ranges from 70 to 80. The value of the average annual air temperature in 1981–2010 in Braniewo poviat was 7.9 °C, which is 0.5 °C higher value than in the years 1971–2000. The average annual sum of precipitation in the years 1981–2010 was 641 mm [55].

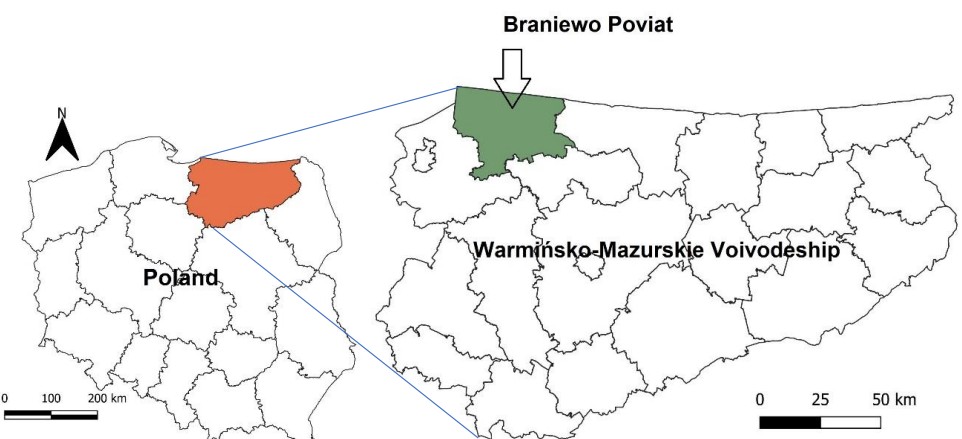

**Figure 1.** Location of the study area.

Pasłęka and Bauda rivers have the largest water resources, measured by the average flow rate [56]. The range of unit surface runoff is close to the average for the Warmińsko–Mazurskie voivodship and ranges from 5 L/s/km$^2$ to 8 L/s/km$^2$ [57]. There are a few natural water reservoirs in the area in question. Only Głębockie Lake with an area of 102.6 ha and a depth not exceeding 9 m can be mentioned here. The largest retention reservoir is Lake Pierzchalskie, created in 1916 by damming the lower section of Pasłęka river with an earth dam. A reservoir with an area of 240 ha and a maximum depth of approximately 10 m is used for energy and flood protection purposes. The total capacity of the reservoir is 11.45 million m$^3$, and the flood capacity is 4 million m$^3$. In Braniewo poviat, there are 11 fishing facilities that retain 1179 thousand m$^3$ of water. The indicated resources should be assessed as small, considering the fact that, in the Warmińsko–Mazurskie voivodship, there are 556 ponds and lakes with a total capacity of 1,028,829 thousand m$^3$ [55].

In Braniewo poviat, there are no in-use valley irrigation facilities of a subsoil nature. Other irrigation systems are used in a small group of farms [55].

The Quaternary aquifer and the Neogene and Palaeogene aquifer are of major utility importance in the area in question. The Quaternary aquifer is formed by sandy water-glacial formations of different stratigraphy, separating boulder clays of different glaciations or occurring on their surface. The Quaternary aquifer occurs in most of Braniewo poviat at a variable depth. In Pasłęka river delta it occurs just below the surface, while in the remaining area at a depth of 20 m to 70 m. Additionally, the thickness of the Quaternary aquifer varies. In the northern and southeastern parts of the district, the thickness is less than 10 m, and in the central part of the district, it is more than 60 m. The conductivity of this aquifer ranges from 50 m$^2 \cdot$d$^{-1}$ to 500 m$^2 \cdot$d$^{-1}$ and the filtration coefficient ranges from 2.5 m$\cdot$d$^{-1}$ to 60 m$\cdot$d$^{-1}$. The groundwater table of the Quaternary aquifer is sub-artesian in some parts of the studied area and free in others [58].

The waters of the main Quaternary aquifer, in relation to the drinking water quality standards, are characterized by a slight excess of permissible values of iron compounds, often also of manganese and in a few cases of ammonia. Therefore, it can be concluded that their quality is suitable for the irrigation of crops in the studied area.

The Neogene level, 10 to 20 m thick, occurs at a depth of 30 to over 60 m. The Paleogene layer is a separate aquifer. It is made of fine-grained and silty, quartz-glauconite sands. The thickness of the aquifer is, on average, 30 to 40 m, and its upper story (ceiling) is at a depth of 120 to 200 m [58].

In Braniewo poviat, approximately 65% of the area is covered by brown and fawn soils made mainly of light loams. Lighter soils, made of clay sands, constitute approximately 8%, while heavy soils made of clay and muds constitute 9% of the poviat's area [59].

According to the data of the Statistical Office in Olsztyn, cereals dominate the structure of crops in Braniewo poviat, which are grown in approximately 90% of farms. Cereals together cover an area of approximately 25.65 thousand ha, which constitutes 74.2% of the

total crops. Other crops are industrial crops, mainly rape and agrimony and root crops such as potato and sugar beet [55].

## 3. Materials and Methods

This paper presents an analysis of the possibilities of covering water shortages in plants most often cultivated in Braniewo poviat with the use of irrigation systems supplied with groundwater. These are the waters present in the entire area covered by the research. Potentially, with varying yields, they can be well captured within each farmland.

The area of crops at risk of water shortages (with a probability of shortages of 20% and 50%) was determined based on the Atlas of water shortages in arable crops and grasslands in Poland [59]. On the other hand, available groundwater resources in the analyzed area were estimated on the basis of the Hydrogeological Map of Poland, prepared by the Polish Geological Institute [58]. For this purpose, a map of the efficiency of a potential drilled well was used, in which the area of Braniewo poviat was divided into six categories in terms of the amount of resources that can be used as a source of irrigation water.

The average-weighted efficiency of a potential well in the poviat was determined from the following relationship:

$$Q_{avg} = \frac{\sum A_i \cdot Q_i}{\sum A_i},\tag{1}$$

where:

$A_i$—total surface area in a given efficiency class [ha],
$A_c$—total area [ha],
$Q_i$—middle of the variation range in class $i$ [m³·h⁻¹].

The percentage shares of the area of individual classes of potential efficiency $\alpha_i$ were determined using the following formula:

$$\alpha_i = \frac{A_i}{A_c} \cdot 100,\tag{2}$$

The comparison of the spatial structure of the occurrence in Braniewo poviat of water shortages of plants and available groundwater resources was carried out using QGIS 3.4 software. For this purpose, the Intersects function from the Vector tab, Geoprocessing Tools, was used.

## 4. Results and Discussion

### 4.1. Climatic Conditions

Air temperature and precipitation are not only the basic elements of the description of climate features but also the meteorological factors that are of greatest importance in agricultural production [60]. Central Europe, including Poland, lies in a temperate climate [10], where the significance of drought is often underestimated [11]. However, with global climate change, the drought problem in the region is becoming increasingly serious [11].

The analysis of the thermal conditions prevailing in the years 1981–2020 shows a clear upward trend in the value of the average annual air temperature, which in this period increased by 1.4 °C (Figure 2). In the case of annual sums of precipitations, a slight increase in the value is noticeable (2 mm·year⁻¹). which was certainly influenced by the situation in 2017, when the annual sum of precipitation was almost twice as high as the norm. The analysis of the precipitation and thermal situation in the region indicates the lack of unidirectional changes in the value of precipitation sums, while maintaining the trend of temperature increase, which is consistent with the observations for the multi-year period 1981–2010, as well as with the results of studies obtained by other authors [61,62].

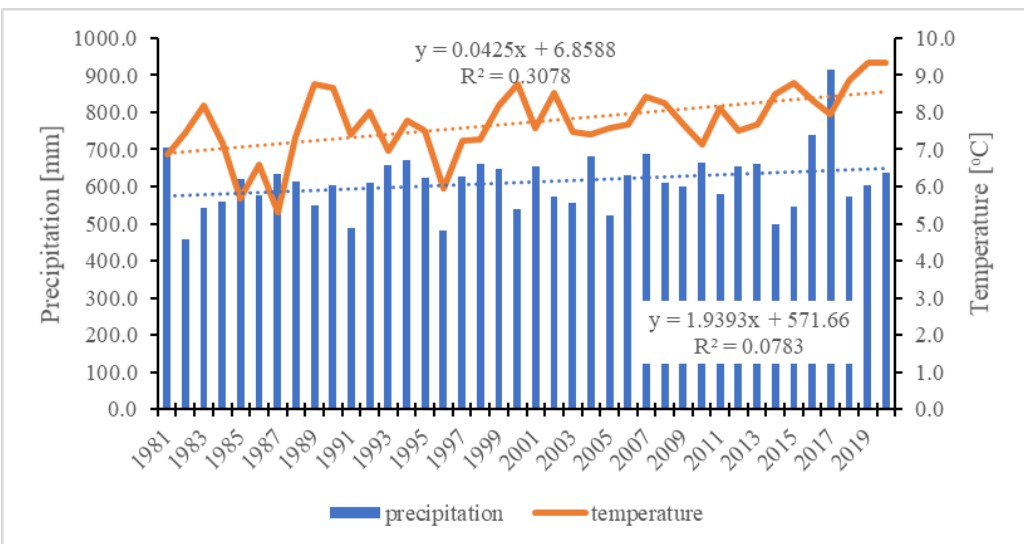

**Figure 2.** The value of the average annual air temperature and sum of precipitation in the years 1981–2020 (area average for Braniewo poviat).

In Braniewo poviat, there was no statistically significant trend of changes in the sum of annual precipitation, but changes in the structure of precipitation were noticed. There was a tendency to decrease the frequency of rainfall in the range of 5.1–10 mm·day$^{-1}$, with a simultaneous increase in daily rainfall in the range from 0.1 mm to 5 mm [55]. The low variability of the sum of precipitation does not mean that the situation related to water resources remains stable. At much higher temperatures, the increase of which is recorded both on an annual and seasonal basis, the intensity of the evapotranspiration process increases, which translates into a reduction in water resources [11]. Moreover, the increase in the temperature value translates into changes in the dates of start, end and duration of thermal seasons and agricultural periods (economic, vegetative and active growth). The duration of early spring and summer was lengthened, and the duration of the other thermal seasons shortened. The trend of changes within 30 years, consisting of the earlier start and later end of the agricultural seasons, has extended the economic period by 20 days, the vegetative period by 13 days and the period of active growth by 6 days [55].

According to the data from the Agricultural Drought Monitoring System, the threat of drought in individual communes of Braniewo poviat occurred in 2009, 2014, 2015, 2018, 2019 and 2020. It has become obvious, especially in recent years (2018, 2019, and 2020), that even the northern part of Central Europe, with its predominantly humid climate, will in the future be threatened by long periods without sufficient rainfall during the summer months [12].

The risk of drought, determined based on climatic water balance, varies depending on the species of crops. Numerous studies have indicated that resistance to water stress is a genotypic trait [63,64]. In plant breeding, changes are made to morphoanatomical, physiological and biochemical changes that will reduce water loss through transpiration and contribute to increasing the efficiency of water use by plants [65]. As has been observed, the risk of drought has occurred 6 times over 11 years, and since 2018, it has been recorded every year. This situation shows that irrigation of agricultural crops has become an important factor in yield in recent years in the discussed area. Irrigation ensures the stability of crop yields, alleviating problems related to the occurrence of droughts caused by climate change [66].

*4.2. Water Needs and Water Shortages of Plants*

Among the plants cultivated in Braniewo poviat, root crops, such as late potato (430–480 mm) and sugar beet (500–550 mm), are characterized by the highest water needs (Table 1). These are crops of key importance to the region's food security. Potato, cultivated

in about 150 countries, with a total acreage of 19 million ha, is one of the most important food crops, after wheat and rice [67]. Compared to other EU countries, Poland is the third largest producer of sugar beet, after France and Germany [66].

**Table 1.** Water needs and maximum water shortages (with a probability of 20% and 50%) in the main crops in Braniewo poviat [mm].

| Crops | Water Demand | Max. Water Shortages $p = 20\%$ | Max. Water Shortages $p = 50\%$ |
|---|---|---|---|
| Rye | 250–280 | 153.3 | 104.6 |
| Winter wheat | 270–300 | 177.5 | 123.2 |
| Spring barley | 360–370 | 167.0 | 109.7 |
| Late potato | 430–480 | 253.3 | 175.4 |
| Sugar beet | 500–550 | 260.6 | 168.3 |
| Winter rape | 350–400 | 66.8 | 34.1 |

Source: ows study based on [59].

It is worth noting that the high sensitivity of potato crops, in particular, to water shortages is due to the shallow and inefficient root system and low regenerative capacity after stress. In the case of sugar beet, it concerns the long vegetation period of this plant and its high yield potential [66].

Every second year, average water shortages may appear in the case of late potato cultivation in the amount of 40–120 mm. However, every five years, there may be an average water shortage in these crops in the amount from 120 mm to 160 mm, covering 89.4% of the crops. They occur mainly in lessive soils made of light loams (45.4%) and in brown soils made of light loams (44%) (Table 2). Water deficiencies affect all stages of potato growth, but the most sensitive stages of development are tuber initiation and the period of intensive growth [68,69]. Climate change models predict an increase in potato yield losses of 32% in the first three decades of this century [70].

**Table 2.** Percentage share of soil type in the area of crops potentially affected by water shortages, with a probability of 20% and 50%, in Braniewo poviat [%].

| Plant | Rye | | Winter Wheat | | | Spring Wheat | | Spring Barley | | Late Potatoes | | | Sugar Beets | | | Winter Rape |
|---|---|---|---|---|---|---|---|---|---|---|---|---|---|---|---|---|
| Range of average shortages (mm) | 0–40 | 40–80 | 0–40 | 40–80 | 80–120 | 0–40 | 40–80 | 0–40 | 40–80 | 40–80 | 80–120 | 120–160 | 40–80 | 80–120 | 120–160 | 0–40 |
| **Water shortage with a probability of 20%** | | | | | | | | | | | | | | | | |
| brown and loamy soils made of clay sands | 10.6 | | | | | | | | | | 10.6 | | | | | 7.6 |
| brown soils made of light loams | | 44.0 | | | 34.4 | | 36.8 | | 36.8 | | | 44.0 | | 34.1 | 4.1 | 31.7 |
| lessive soils made of light loams | | 45.4 | | | 35.5 | | 38.0 | | 38.0 | | | 45.4 | | 30.2 | 9.4 | 32.8 |
| brown soils made of medium loams | | | | 2.1 | 17.8 | | 21.3 | | 21.3 | | | | | 22.2 | | 18.4 |
| brown and lessive soils made of clay | | | | 6.5 | | | | | | | | | | | | 6.0 |
| medium and heavy sacks | | | | 3.6 | | 3.9 | | 3.9 | | | | | | | | 3.4 |
| **Water shortage with a probability of 50%** | | | | | | | | | | | | | | | | |
| brown and loamy soils made of clay sands | 10.6 | | | | | | | | | 10.6 | | | | | | 7.6 |
| brown soils made of light loams | | 44.0 | | 34.4 | | | 36.8 | | 36.8 | 28.0 | 16.0 | | 38.3 | | | 31.7 |
| lessive soils made of light loams | | 45.4 | | 35.5 | | | 38.0 | | 38.0 | 33.9 | 11.5 | | 39.5 | | | 32.8 |
| brown soils made of medium loams | | | | 19.9 | | | 21.3 | | 21.3 | | | | 22.2 | | | 18.4 |
| brown and lessive soils made of clay | | | | 6.5 | | | | | | | | | | | | 6.0 |
| medium and heavy sacks | | | | 3.6 | | 3.9 | | 3.9 | | | | | | | | 3.4 |

Source: Own study based on [59].

Water shortages in the case of sugar beet cultivation, with the probability of occurrence above 50%, are usually in the range of 40–80 mm, and with the probability of occurrence of 20% and more, mainly in the range of 80–120 mm (Table 2). The area of occurrence of these deficiencies covers 86.5% of the crops located on brown soils made of light and medium

loams and on lessive soils made of light loams. In a small area, where there are mainly light clay soils, there are higher deficiencies (on average, in the range of 120 mm to 160 mm).

It is estimated that the water deficit in the growing season, the frequency of which is around 30% [71], causes a drop in sugar beet yields in Europe by 5% to 30% annually [72]. This trend is increasing in dry and semi-dry regions [73]. In order to obtain higher yields, full-dose irrigation of a complementary and intervention nature is necessary. Vamerali et al. [74] argued that sugar beet is also a field crop well suited for deficit irrigation. The sugar beet yield in years with insufficient water quantity does decrease, but the sugar content in tubers significantly increases [60].

Wheat, on the other hand, is one of the major food crops in the world [75], and wheat-based food is a major component of the human diet [76]. Water scarcity affects the growth of wheat at various stages of development. The impact of water stress on the photosynthesis process in cereals, especially wheat, depends on the duration and intensity of stress [5]. The limitation of photosynthesis consequently leads to a decrease in biomass (even by $\frac{1}{4}$) [77] and the yield of plants [5]. Winter wheat sowing in Braniewo poviat occurs on all types of soil except for the lightest soils (brown and lessive made of clay sands). Every fifth year, water shortages may occur on average in the amount of 80–120 mm on 87.7% of crops, and once every two years in the amount of 40–80 mm on 89.8% of crops.

As shown by extensive research conducted in the Carpathian Basin, the irrigation of wheat is not profitable due to the high energy costs. Therefore, the basic solution for minimizing yield losses may be the effective use of available water resources from the soil [78,79].

The smallest water shortages in Braniewo poviat concern the cultivation of rye, spring wheat, spring barley and winter rape. Winter oilseed rape is an important oilseed plant in Europe that is grown mainly for cooking oil and biofuels. In the European Union, the highest rape yields are recorded in France, Germany and Poland [80].

The conducted research has shown that in Braniewo poviat, water shortages not exceeding 40 mm on average may occur every 2 years, and with a probability of 20% (except for rape), in the case of approximately 90% of crops, shortages in the amount of 40–80 mm may occur. The natural drought tolerance of barley is related to the early flowering of this plant, which ensures optimal pollination, seed development and maturation in the optimal period [9,81].

*4.3. Available Groundwater Resources*

Irrigation with the use of groundwater resources present in the analyzed area can be a way to cover water shortages in agricultural crops. The resources of these waters were estimated on the basis of the data compiled on the Hydrogeological Map of Poland [58], and their distribution is shown in Figure 3.

On 41.19% of the area of Braniewo poviat, there is groundwater with the potential capacity of a drilled well in the range of 30–50 $m^3 \cdot h^{-1}$. The areas with potential capacities lower than the average (below 30 $m^3 \cdot h^{-1}$) occur along the western border of the poviat and in the middle belt, from the village of Wyszkowo (in Lelkowo commune) toward the west along the border between the communes of Braniewo, Pieniężno and Płoskinia (Figure 3). They cover a total of 27.44% of the poviat area. In a similar area (31.37% of the poviat's area), there are areas richer in groundwater that can be used for irrigation of agricultural crops. The average capacity of a potential well in the poviat is 42.4 $m^3 \cdot h^{-1}$ (Table 3).

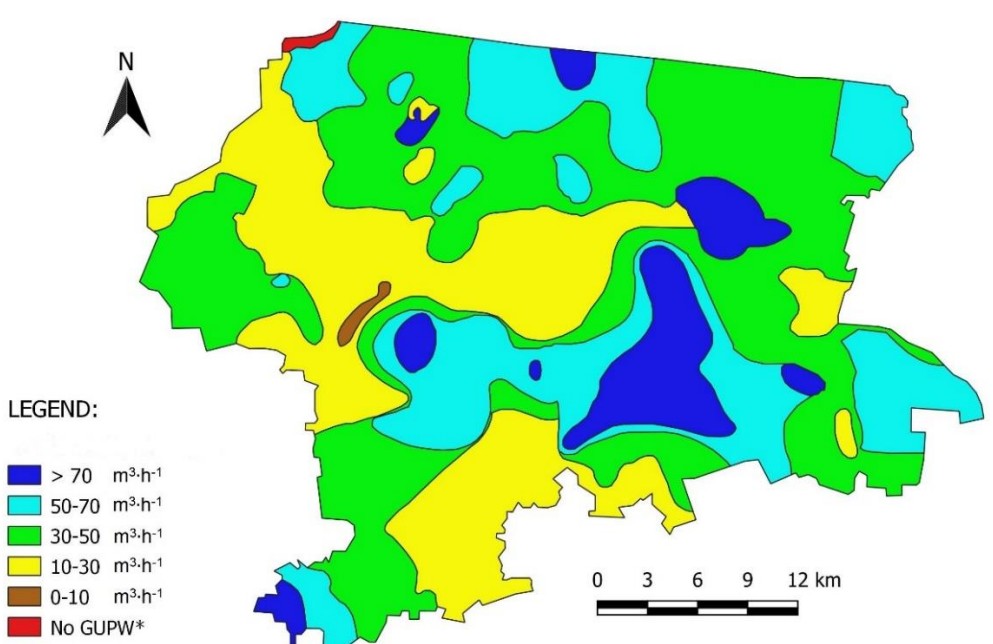

**Figure 3.** Areas with the potential capacity of a drilled well of main utility level in Braniewo poviat (own study based on [58]). * GUPW—the main usable groundwater level.

**Table 3.** Characteristics of groundwater resources in Braniewo poviat based on the analysis of the potential capacity of a drilled well containing the main usable groundwater level.

| Potential Well Performance Class i | Potential Capacity Value Range in a Given Class $Qmin_i \div Qmax_i$ ($m^3 \cdot h^{-1}$) | The Middle of the Range of Variability $Q_i$ ($m^3 \cdot h^{-1}$) | Number of Areas of a Given Class Within a Poviat $N_i$ | The Total Area in a Given Efficiency Class $A_i$ (ha) | Percentage Share of the Efficiency Class in the Poviat Area $\alpha_i$ (%) | Weighted Average Capacity of Potential Wells in the Poviat Area $Q_{avg}$ ($m^3 \cdot h^{-1}$) |
|---|---|---|---|---|---|---|
| 1 | No GUPW * | 0 | 1 | 228.25 | 0.19 | |
| 2 | 0–10 | 5 | 1 | 320.89 | 0.27 | |
| 3 | 10–30 | 20 | 8 | 32,496.17 | 26.98 | |
| 4 | 30–50 | 40 | 3 | 49,615.51 | 41.19 | 42.40 |
| 5 | 50–70 | 60 | 10 | 27,621.01 | 22.93 | |
| 6 | >70 | 80 | 8 | 10,171.47 | 8.44 | |
| Sum | | | 31 | 120,453.30 | 100.00 | |

* GUPW—the main usable groundwater level. Source: ows study based on [59].

*4.4. Possibility of Covering Potential Water Shortages in Plants*

The comparison of the areas of water shortage occurrence of individual crops with the capacity of a potential drilled well collecting the main usable groundwater level allows us to determine whether there are possibilities to cover the shortages with irrigation systems supplied from groundwater wells.

The conducted analysis showed that the largest part of agricultural crops in Braniewo poviat are located in areas with an average abundance of available groundwater (Tables 4 and 5). In these areas, the greatest water shortages, with a probability of 20%, reaching up to 160 mm, are related to the cultivation of sugar beet and late potatoes.

**Table 4.** Percentage of the area of crops potentially affected by water shortages, with a probability of 50% in the area of groundwater resources in the distinguished performance ranges of a drilled well [%].

| Crop | Size of Shortages | The Range of Potential Capacity of a Drilled Well Entering the Main Usable Groundwater Level | | | | | |
|------|------|------|------|------|------|------|------|
| | (mm) | 0 | 0–10 | 10–30 | 30–50 | 50–70 | >70 |
| Rye | 0–40 | 0.0 | 0.0 | 11.4 | 33.5 | 18.7 | 8.7 |
| Winter wheat | 0–40 | 0.0 | 0.3 | 4.0 | 3.2 | 1.3 | 0.5 |
| | 40–80 | 0.0 | 0.0 | 20.6 | 36.2 | 18.2 | 8.4 |
| Spring wheat | 0–40 | 0.0 | 0.0 | 21.3 | 38.3 | 18.5 | 8.6 |
| Spring barley | 0–40 | 0.0 | 0.0 | 21.3 | 38.3 | 18.2 | 8.6 |
| Late potato | 40–80 | 0.0 | 0.0 | 6.1 | 23.4 | 15.1 | 7.8 |
| | 80–120 | 0.0 | 0.0 | 5.4 | 10.0 | 3.5 | 1.0 |
| Sugar beet | 40–80 | 0.0 | 0.0 | 20.7 | 36.1 | 17.8 | 8.4 |
| Winter rape | 0–40 | 0.0 | 0.3 | 24.9 | 43.2 | 22.3 | 9.3 |

Source: own study based on [58,59].

**Table 5.** Percentage of the area of crops potentially affected by water shortages, with a probability of 20% in the area of groundwater resources in the distinguished performance ranges of a drilled well (%).

| Crop | Size of Shortages | The Range of Potential Capacity of a Drilled Well Entering the Main Usable Groundwater Level | | | | | |
|------|------|------|------|------|------|------|------|
| | (mm) | 0 | 0–10 | 10–30 | 30–50 | 50–70 | >70 |
| Rye | 0–40 | 0.0 | 0.0 | 0.2 | 3.9 | 3.1 | 0.4 |
| | 40–80 | 0.0 | 0.0 | 11.2 | 29.5 | 15.5 | 8.4 |
| Winter wheat | 40–80 | 0.0 | 0.3 | 6.0 | 3.2 | 1.3 | 0.5 |
| | 80–120 | 0.0 | 0.0 | 18.7 | 36.1 | 17.8 | 8.4 |
| Spring wheat | 0–40 | 0.0 | 0.0 | 0.7 | 2.1 | 0.4 | 0.2 |
| | 40–80 | 0.0 | 0.0 | 20.6 | 36.2 | 17.8 | 8.4 |
| Spring barley | 0–40 | 0.0 | 0.0 | 0.7 | 2.1 | 0.4 | 0.2 |
| | 40–80 | 0.0 | 0.0 | 20.7 | 36.1 | 17.8 | 8.4 |
| Late potato | 80–120 | 0.0 | 0.0 | 0.2 | 3.9 | 3.1 | 0.4 |
| | 120–160 | 0.0 | 0.0 | 11.2 | 29.6 | 15.5 | 8.4 |
| Sugar beet | 80–120 | 0.0 | 0.0 | 16.6 | 29.2 | 14.7 | 7.5 |
| | 120–160 | 0.0 | 0.0 | 4.0 | 7.0 | 3.2 | 0.9 |
| Winter rape | 0–40 | 0.0 | 0.3 | 24.9 | 43.2 | 22.3 | 9.3 |

Source: own study based on [58,59].

About 11% of rye and late potatoes, or about 21–25% of wheat, spring barley, sugar beet or winter rape may be cultivated in areas with a capacity below the average potential drilled well entering the main usable groundwater level. Better conditions for the application of irrigation occur in the area of 26–31% of the crops, with 8–9% located in the areas richest in groundwater. The remaining areas, which constitute 100% of the values given in the tables, are areas not used for agriculture (forests, waters, urban areas), where soil conditions are not suitable for cultivating selected crops. The above-mentioned proportions take into account potential water shortages with a probability of 50% and 20%, with those that may occur every five years having higher values. The exception to this is rape crops, for which the size of shortages with a probability of 50% and 20% and their spatial extent turned out to be similar.

In Braniewo poviat, representing the conditions of Central and Eastern Europe, there are average possibilities for irrigating crops that use groundwater resources as a source of water. Currently, irrigation in the poviat is used on a small number of farms and to a limited extent. However, in the context of climate change and increasingly frequent droughts, agriculture is predicted to face major challenges, as it will have to feed a growing

population with diminishing water resources [31,41]. In many countries, appropriate acts were adopted, measures were also taken to improve the capacity of institutions responsible for managing water resources, and practices aimed at adapting agriculture to climate change were implemented [21].

## 5. Conclusions

When planning the implementation of irrigation systems that require larger amounts of water, the possibility of using infiltration intakes should be considered. These are wells collecting water from the first aquifer, located in the vicinity of infiltrating surface waters (watercourses, natural and artificial reservoirs, wetlands, with the exception of ecological sites, etc.). This source of water, shallower below the surface of the terrain, is faster renewable, and the range of the depression cone in the case of such an intake is smaller.

The greatest water needs are found in root crops grown in the poviat, including late varieties of potato (430–480 mm) and sugar beet (500–550 mm). Additionally, water shortages in these crops are the highest. On average, they can be up to 160 mm with a probability of 20% and up to 120 mm with a probability of 50%. This clearly indicates the need for irrigation of agricultural crops. Groundwater is the most common source of water in the area for supplying irrigation systems. It was found that good conditions for the irrigation of agricultural crops in Braniewo poviat with groundwater occur on 26–31% of the cultivated area, with 8–9% being the most abundant in groundwater. Most of the crops are located in the area of medium abundance in available groundwater (30–50 $m^3 \cdot h^{-1}$). One well of this discharge rate will cover water shortages in late potato or sugar beet crops in the area of 11.5–34.5 ha. About 11% of rye and late potato crops and about 21–25% of wheat, spring barley, sugar beet or winter rape are located in areas with lower well capacity (less than 30 $m^3 \cdot h^{-1}$).

The intensive use of groundwater can have negative and irreversible effects on the environment. The renewal time of groundwater resources is much longer than that of surface waters. Therefore, their use for irrigation should be limited to the necessary minimum (mainly to supply systems that require small amounts of water—micro-irrigation plants, drip irrigation) and to areas where there are no other sources of water.

**Author Contributions:** Conceptualization, I.C. and E.D.; methodology, I.C.; software, I.C.; validation, I.C., E.D. and Z.B.; formal analysis, I.C.; investigation, I.C., E.D. and Z.B.; resources, I.C., E.D. and Z.B.; data curation, I.C., E.D. and Z.B.; writing—original draft preparation, I.C. and E.D.; writing—review and editing, I.C., E.D. and Z.B.; visualization, I.C.; supervision, I.C.; project administration, I.C.; funding acquisition, I.C., E.D. and Z.B. All authors have read and agreed to the published version of the manuscript.

**Funding:** The results presented in this paper were obtained as part of a comprehensive study financed by the University of Warmia and Mazury in Olsztyn, Faculty of Agriculture and Forestry, Department of Water Management and Climatology (grant No. 30.610.008–110). Project financially supported by Minister of Education and Science in the range of the program entitled "Regional Initiative of Excellence" for the years 2019–2022, Project No. 010/RID/2018/19, amount of funding 12,000,000 PLN.

**Institutional Review Board Statement:** Not applicable.

**Informed Consent Statement:** Not applicable.

**Data Availability Statement:** Not applicable.

**Conflicts of Interest:** The authors declare no conflict of interest.

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
