# Peer review of "Potential Possibilities of Using Groundwater for Crop Irrigation in the Context of Climate Change"

_agriculture, doi:10.3390/agriculture12060739_

Round 1

Reviewer 1 Report

Comments to the Authors

The entitled manuscript of Using groundwater for irrigation of agricultural crops in the context of climate change is organized and written well and provides valuable information about the structure of the occurrence of available groundwater resources and on this basis to assess the possibility of covering the water shortages in cultivated plants with these resources.

Specific comment:

Figure 2: define both y-axes and remove the equations into the clear part of the figure so it will not be blocked by the bars.

Reviewer 2 Report

Title

The title is abording climate change approach however, the authors only mentioned air temperature and precipitation. The objective of the paper was mentioned to find an alternative (groundwater) to irrigate crops. A recommendation is to modify the title to something related to the effects of using groundwater linked with those aspects of climate change they aborded.

Introduction

The authors mentioned interesting data about the global situation and some scenarios. It could be a strong introduction if they add information about IPCC's recent reports, Sustainable Development Goals (SDG), and the scenarios in the country and study area.

Lines 69 onwards

The authors analyzed the irrigation situation, pros, and cons, but even when they mentioned, it at the end of the paper, it appears that the final idea is to maintain and/or increase drip irrigation systems as the solution. I do not know if the crops are property of big farmers or companies or if most are familiar farms. Drip irrigation systems are sustainable for small farms? Authors mentioned the main problem is the lack of investments in irrigation systems, however, is there another solution, something as Solutions based on Nature (SbN), etc., with low cost, replicable in small communities, with less necessity of electricity, etc., to adapt and to mitigate climate change?

Lines 105 onwards

It is mentioned special care to irrigate crops with groundwater and there are some examples of Mediterranean countries due to salinity and soil degradation. It is necessary to detail why groundwater irrigation is a problem, which processes are linked to this, and if is possible, examples in the study area or country, not just Mediterranean countries. 

Study area

There is not a section about the hydrogeology of the aquifer. This is necessary to understand the nature and characteristics of the disponibility of water and the quality too. The objective of the paper can be enriched if the authors provide hydrogeochemical information. Do the authors know if the quality of the groundwater in the aquifer is adequate for the crops in the study area?

Results and discussion

Is recommended a better edition in Figure 2, equations are difficult to read without a zoom. Please add “Y axis” title too.

4.2 Water needs and water shortages of plants

Authors described late potato and sugar beet as very important crops for food security in the region. However, there are bigger crops and economically important crops like wheat and rice. If food security is a key aspect to maintain late potato and sugar beet, could be interesting more data about the goodness of these crops over wheat and rice, especially in the context of mitigation and adaptation to climate change and the SDG.

Adaptation to the new temperature and precipitation conditions in the study area is critical to maintaining crops. Do the Authors know about some local farmer strategies implemented in this region or in other similar regions? Even when this is not the objective in the paper, due to the high cost of the irrigation network, energy supply, and carbon print, is adequate an alternative or complementary plan.

4.3 Available groundwater resources

The authors determined an available groundwater source for the region. Groundwater quality is adequate for these crops? How many populations live in the area? Do the authors know if the quantity of water in the aquifer is enough to maintain crops and population?

Figure 3. Please add units in the legend.

4.4 Possibility to cover potential water shortages in plants.

Are there other water concessions in the information of the aquifer?

Conclusions

The authors said “Intensive use of groundwater can have negative and irreversible effects on the environment”, but in the text is mentioned groundwater use for irrigation need special care. How much is the intensive use of groundwater? Is it just the quantity or is important the groundwater chemical composition?

Reviewer 3 Report

The manuscript submitted by Cymes et al. entitled "Using groundwater for irrigation of agricultural crops in the context of climate change" deals on the effects of climate change on water supply and crop water efficiency in Poland. This study, which is purely bibliographical in its first part, in its second part tries to evaluate the possibility of supporting crops with the help of groundwater. Unfortunately, the sustainability of this system has not been fully analysed and I hope it will be carried out in the studies. The quality of the manuscript, with the exception of some figures in the results section and some typos in the units, is good in line with the general organisation.

In general, the experimental activity was carried out following a strict scientific logic by following a simple and clear calculation procedure and the quality of the manuscript, with the exception of some figures in the results section and some typos in the units, is well presented the scientific framework, materials and methods and results in a clear manner.
The Abstract is exhaustive and well represents the experimental activity conducted.
Keywords: are appropriate.
Introduction is well written and articulated clearly presents the need for a more efficient agricultural use of water.
Material and Methods: The study area was clearly identified and thoroughly described. However, more details can be added regarding the QGIS analysis.
Results and Discussion: Figure 2 should be revised. See also the unit at line 198. Results were clearly presented and in each section the discussion of the data was adequate and accurate. Le water requirements of the analyzed crops were well presented and discussed.
Conclusions: Are good. However, I suggest moving the first paragraph to the end. The conclusions are based on the achieved results and support the possibility to use groundwater to supply adequate water to the major crop cultivated in the experimental area.
